# One-step propylene purification from a quaternary mixture by a single physisorbent

Peixin Zhang [1,2], Zhensong Qiu[1], Yechen Liu[1], Sen Chen[1,2], Lifeng Yang [1,3] ✉, Xian Suo [2], Xili Cui [1,2] & Huabin Xing [1,2] ✉

One-step removal of multiple impurities implemented by adsorptive separation is an efficient and simple process to afford high purity products, but is hindered by the lack of advanced porous materials that could capture different types of molecules. Herein, a series of novel metal-organic frameworks ZU-921 to ZU-924 with cooperative binding environment of integrated aromaticity surface and fluorine/oxygen electronegative sites are designed, and ZU-921 is presented as the demonstration that solves the long-standing challenge in one-step propylene ($C_3H_6$) purification from the C3 quaternary mixture. The selective recognition ability towards alkyne, allene, alkane than alkene implemented by ZU-921 is attributed to the optimal interaction contribution from polarizability and dipole/quadruple moments that is realized by the fine-tuned density of parallelly-distributed electronegative sites via ligand engineering strategy. Ultra-high purity (99.99%) $C_3H_6$ could be directly obtained from the C3 quaternary mixture ($C_3H_4/C_3H_4(PD)/C_3H_8/C_3H_6$ 1 v/1 v/3 v/95 v) with the productivity of 17.27 L/kg derived from the 10-times scale-up column (1.0 cm × 50 cm) breakthrough experiment. This work not only presents a common strategy in advanced adsorbents design for multiple impurities capture but also provides an energy-efficient alternative for $C_3H_6$ purification.

Multiple impurity capture is the essential step in gas purification, determining the gas purity, and is closely associated with the energy cost of the process[1–5]. Polymer-grade (>99.5%) propylene ($C_3H_6$) is the critical bulk commodity, and its purification involves the removal of complex impurities with similar properties, like propyne ($C_3H_4$, ~1%), propadiene ($C_3H_4$ (PD), ~1%), and propane ($C_3H_8$, ~3%)[6,7]. Currently, the purification of alkene relies on the tandem separation process in industry, including catalytic hydrogenation (noble-metal catalyst at high temperature and pressure) and cryogenic distillation (over 100 trays at 243 K and 0.3 MPa). The energy-intensive nature of the above separation process has spurred research into the development of nonthermal-driven separation technology[8,9].

Adsorptive separation is recognized as a potentially energy-saving alternative to solve challenging separations, and considerable achievements have been gained along with the continuous development of advanced porous materials[10,11], like metal-organic frameworks (MOFs)[12–17], covalent-organic frameworks (COFs)[18,19], etc.[20–23]. Benefiting from their demonstrated fine-tuning ability of pore size and pore chemistry, tailor-made porous materials towards $C_3H_6$ purification have been developed, such as binary mixtures of $C_3H_4/C_3H_6$[24,25], and $C_3H_6/C_3H_8$[26–28], ternary mixtures of $C_3H_4/C_3H_4(PD)/C_3H_6$[29,30]. In detail, the designed porous materials, decorating the pore surface with polar groups, like anions[31], open metal sites[32], are able to preferentially adsorb the molecules with higher dipole/quadrupole moments, and remove trace alkyne from $C_3H_6$ mixtures, as well as the selective capture of $C_3H_6$ from $C_3H_8$. Constructing an alkane trap to introduce high-density weak interaction sites can enhance the binding affinity with $C_3H_8$, and realize the selective capture of $C_3H_8$ from $C_3H_6$ mixtures[33,34],

[1]Zhejiang Key Laboratory of Intelligent Manufacturing for Functional Chemicals, College of Chemical and Biological Engineering, Zhejiang University, Hangzhou, China. [2]Engineering Research Center of Functional Materials Intelligent Manufacturing of Zhejiang Province, Institute for Intelligent Bio/Chem Manufacturing, ZJU-Hangzhou Global Scientific and Technological Innovation Center, Hangzhou, China. [3]State Key Laboratory of Silicon Materials, School of Materials Science and Engineering, Zhejiang University, Hangzhou, China. ✉e-mail: lifeng_yang@zju.edu.cn; xinghb@zju.edu.cn

but fail to simultaneously capture the $C_3H_4$, $C_3H_4$(PD) with lower polarizability. Controlling pore size to create molecular sieves that could exclude molecules of large size, achieving the separation of $C_3H_4$ and $C_3H_6$ from $C_3H_4/C_3H_6$ and $C_3H_6/C_3H_8$ mixtures, respectively[35–38]. However, despite the above progress, one-step $C_3H_6$ purification from quaternary C3 mixtures containing $C_3H_4$, $C_3H_4$(PD), and $C_3H_8$ via a single physisorbent still remains a grand challenge[39].

To realize one-step $C_3H_6$ purification, the adsorbents are expected to show higher binding affinity towards all C3 alkyne, allene, and alkane than alkene. However, as revealed by the physiochemical properties of the four gases, they have very similar properties, especially for $C_3H_6$ and $C_3H_8$. The polarizability, dipole/quadrupole moments, and molecular size of $C_3H_6$ all lie between $C_3H_4/C_3H_4$ (PD) and $C_3H_8$. Meanwhile, $C_3H_8$ shows the highest polarizability but the lowest dipole/quadrupole moments, while the condition is reversed for $C_3H_4$ and $C_3H_4$(PD). The fact indicates that the different binding sites or environments and their fine-tuning are required to realize the preferential accommodation of $C_3H_4$, $C_3H_4$(PD), and $C_3H_8$ than that of $C_3H_6$ in the confined channel, posing a daunting challenge to the design of porous materials (Fig. 1a and Fig. S1). In detail, to fulfill the target of simultaneous capture of $C_3H_4$, $C_3H_4$ (PD), and $C_3H_8$, the design of porous materials needs to overcome two great challenges. (1) Creating a molecular trap that shows preferential adsorption towards $C_3H_8$ than $C_3H_6$, the small polarizability difference between $C_3H_8$ and $C_3H_6$ makes it challenging (polarizability: $C_3H_8$ $62.9$–$63.7 \times 10^{-25}$ cm$^3$ vs $C_3H_6$ $62.6 \times 10^{-25}$ cm$^3$)[33,34,40], the $C_3H_8/C_3H_6$ selectivity of most reported $C_3H_8$-selective materials is around $1.5$[41,42]. (2) Creating the cooperative binding environment that recognizes molecules via both polarizability and dipole/quadrupole moments[43,44]. Different affinity sequences of the four C3 gases could be realized via fine-tuning the interaction contribution from the different kinds of binding sites[32] (Fig. 1b, c). Through the exploitation of more H interaction sites of $C_3H_8$ than $C_3H_6$ and higher electropositivity H atoms of $C_3H_4$, $C_3H_4$(PD) than $C_3H_6$,

porous materials with an optimal cooperative binding environment rationally show the weakest $C_3H_6$ affinity. Following this idea, the fluorine/oxygen polar binding sites are integrated into the aromatic-based alkane trap that is mainly dominated by the dispersion/induction interactions based on polarizability.

Through controlling the density of parallely distributed fluorine via a ligand engineering strategy, the contribution of interactions from the dipole/quadrupole moments and polarizability could be fine-tuned. Herein, we solved the challenge of one-step $C_3H_6$ purification from the quaternary mixtures using tailor-made ZU-921 (ZU represents Zhejiang University) featuring orderly lined aromatic surfaces and optimal density of electronegative sites. The selectively adsorbed alkyne, allene, and alkane over alkene molecules are bonded simultaneously via hydrogen-bonding interactions and multiple van der Waals interactions with high selectivity of alkane/alkene (2.03), alkyne/alkene (2.17), and allene/alkene (2.03). High-purity (99.99%) $C_3H_6$ could be directly obtained from C3 quaternary mixtures ($C_3H_4/C_3H_4$(PD)/$C_3H_8$/ $C_3H_6$ 1 v/1 v/3 v/95) with the productivity of around 17.27 L/kg using ZU-921 as demonstrated by the 10-times scale-up column breakthrough experiments. The molecular-level understanding of the adsorption behavior for C3 gases within the well-defined pore space highlighted the importance of the rational integration of synergistic binding sites to enhance the recognition ability of multiple gases with different properties.

## Results

### Synthesis and characterization

A series of isostructural ultramicroporous materials, ZU-921 ([Co(IPA-F)(DPG)]$_n$, DPG = meso-α, β-di(4-pyridyl) glycol, IPA-F = 5-fluoroisophthalic acid), ZU-922 ([Co(IPA-CH$_3$)(DPG)]$_n$, IPA-CH$_3$ = 5-methylisophthalic acid), ZU-923 ([Co(BDC-2F)(DPG)]$_n$, BDC-2F = 2,5-difluoroterephthalic acid) and ZU-924 ([Co(BDC-4F) (DPG)]$_n$, BDC-4F = tetrafluoroterephthalic acid) were successfully

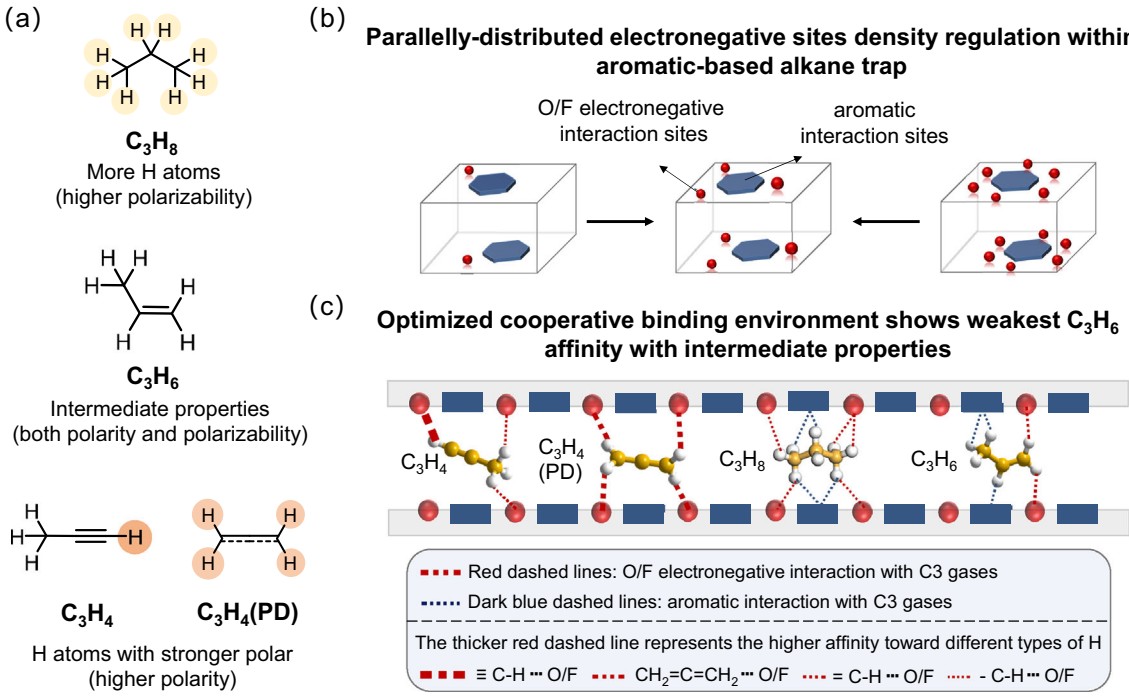

**Fig. 1 | Schematic diagram of the adsorbent design and adsorption behavior for one-step $C_3H_6$ purification. a** The properties difference of $C_3H_4$, $C_3H_4$ (PD), $C_3H_6$, and $C_3H_8$, **b** the illustration of the strategy of adsorbents design. Introducing O/F electronegative sites into the aromatic-based propane trap to form a cooperative binding environment, and controlling the density of introduced electronegative sites to fine-tune the binding affinity sequence of the four C3 gases, **c** the adsorption behavior of $C_3H_4$, $C_3H_4$ (PD), $C_3H_6$, and $C_3H_8$ under the optimal cooperative interaction environments. $C_3H_4$ and $C_3H_4$ (PD) exhibit higher affinity with O/F than $C_3H_6$ due to their higher polarity, while $C_3H_8$ could form more dense interactions with the aromatic-based sites and O/F sites than $C_3H_6$.

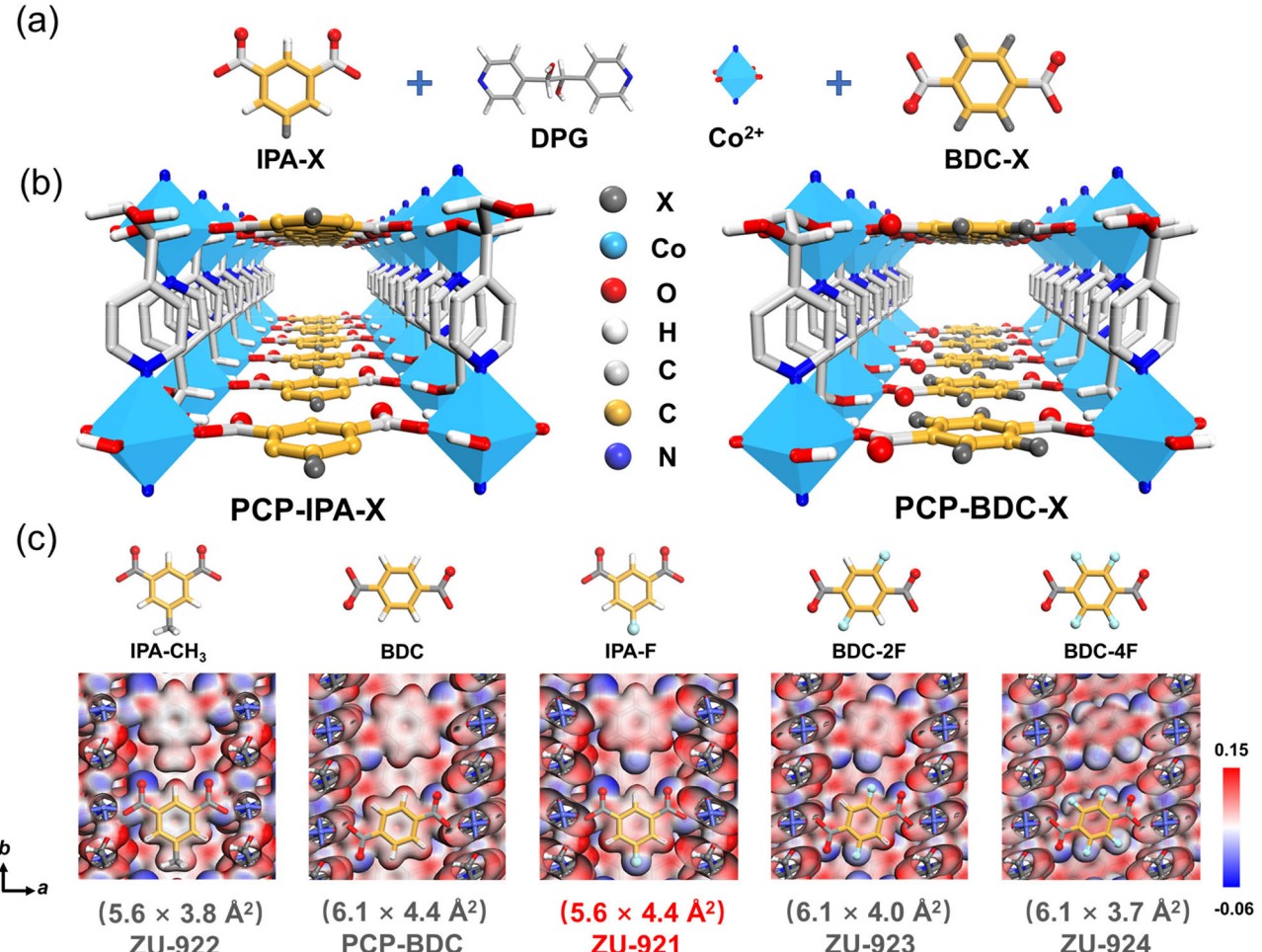

**Fig. 2 | Scheme and structure of five isostructural MOFs. a** The building blocks (Co²⁺, DPG, and organic ligand of IPA-X and BDC-X, X represents different functional groups), **b** the structure of the three-dimensional framework of PCP-IPA-X and PCP-BDC-X, and **c** the electrostatic potential maps of ZU-921 to ZU-924 and PCP-BDC and their pore size.

synthesized through solvothermal reactions of Co(NO₃)₂·6H₂O, meso-α, β-di(4-pyridyl) glycol and their corresponding dicarboxylate acid. The structures of ZU-921 to ZU-924 are isostructural except for the functional groups, as revealed by their similar powder X-ray diffraction (PXRD) patterns (Fig. S2). The method of Rietveld refinement was adopted to obtain the refinement structures of ZU-921 and ZU-922 based on the parent structure of [Co(IPA)(DPG)]ₙ (IPA = isophthalic acid)[45], and the low $R_p$ (0.0143, 0.0142) and $R_{wp}$ (0.0298, 0.0270) values indicate the high quality of the analyzed structures (Fig. S3, Table S2 and "Structure simulation" section). The PXRD patterns of the synthesized power of ZU-921 and ZU-922 are well matched with the simulated one of the refined structures (Fig. S4). Individually, each Co(II) atom was connected by two pyridine groups and two hydroxyl groups from independent meso-α, β-di(4-pyridyl) glycol ligands to form a 2D layer network, and the 2D layer network was further pillared by the dicarboxylate acid (IPA-F and IPA-CH₃, BDC, BDC-2F and BDC-4F) to afford a 3D framework with one-dimensional straight channel (Fig. 2a, b). The pore windows of ZU-921 to ZU-924 are estimated to be 5.6 × 4.4 Å², 5.6 × 3.8 Å², 6.1 × 4.0 Å², and 6.1 × 3.7 Å² by the model of Connolly surface with probe of diameter 1.0 Å (Fig. 2b)[46]. The channels of these isostructural materials are featured with the parallel-aligned linearly extending isophthalic acid units, which can provide a big π system and

multiple hydrogen bond acceptors for the accommodation of alkane. Through ligand engineering strategy, the surface electrostatic potential of the pore environment could be well fine-tuned via controlling the types and density of functional groups. We could observe that the pore channel shows more negative electrostatic potential with the increased density of polar fluorine functional sites, which would enhance the binding affinity with polar molecules (Fig. 2c). The permanent porosity of the synthesized porous materials was investigated by 77 K N₂ and 195 K CO₂ adsorption-desorption isotherms (Figs. S5 and S6). The corresponding surface area and pore volume were calculated and summarized in Table S3. The Langmuir surface area and pore volume were determined as 356.6 m² g⁻¹ and 0.145 cm³ g⁻¹ for ZU-921, and 369.5 m² g⁻¹ and 0.150 cm³ g⁻¹ for ZU-922, respectively. The pore size distribution (PSD) of ZU-921 and ZU-922 is centered at 4.9 Å and 4.5 Å, respectively, which agrees well with the theoretical pore size from crystal simulation (Figs. 1b and S6). Moreover, thermogravimetric analysis (TGA) curves demonstrated the good thermal stability of ZU-921 and ZU-922, and their decomposition temperatures are up to 280 °C and 300 °C, respectively (Fig. S7). The morphology was investigated using a NOVA 200 Nanolab scanning electron microscope (SEM) (Fig. S8). Additionally, the invariable PXRD patterns and BET results indicate that ZU-921 is stable in different solvents

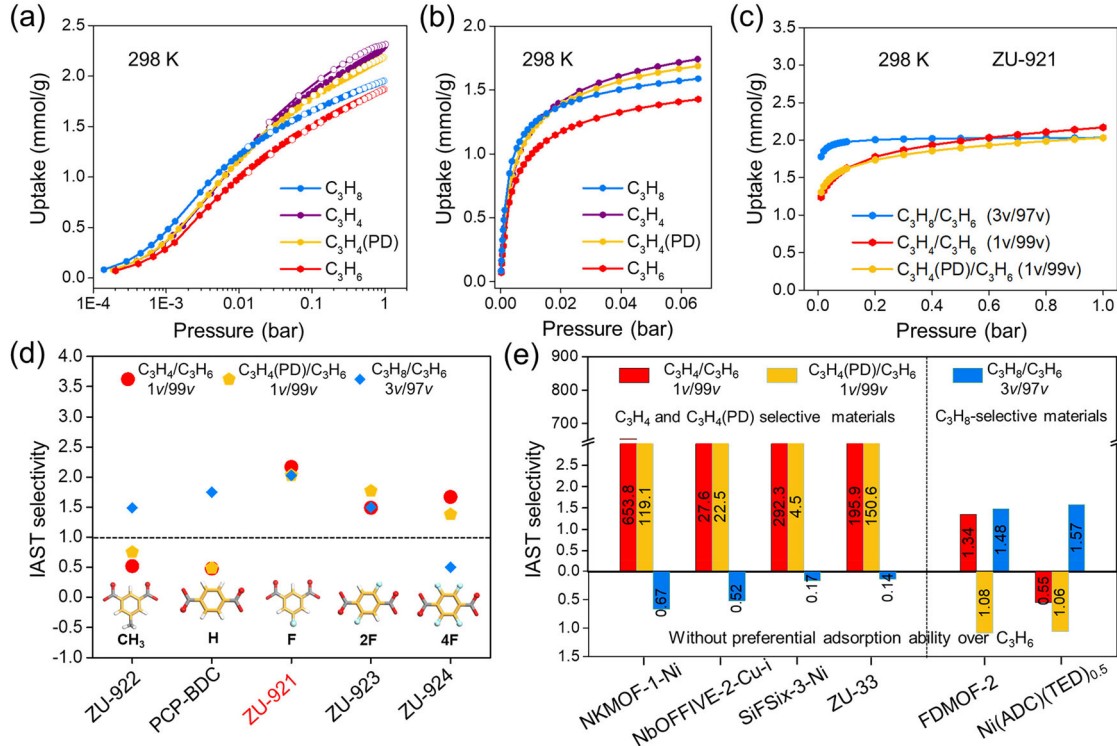

**Fig. 3 | Adsorption and separation performance.** The $C_3H_4$, $C_3H_4$(PD), $C_3H_8$, and $C_3H_6$ adsorption isotherms of ZU-921 at 298 K with (**a**) a logarithm scale under the pressure range of 0–1.0 bar, (**b**) a linear scale under the pressure range of 0–0.06 bar, **c** the IAST selectivity of C3 binary mixtures on ZU-921, **d** comparison of the IAST selectivity of C3 binary mixtures for the series of ZU-921 to ZU-924 and PCP-BDC at 298 K, and **e** comparison of the IAST selectivity of different C3 binary mixtures on ZU-921 with reported benchmark materials for C3 separation. Source data are provided as a Source Data file.

(MeOH, EtOH, MeCN, Acetone, and DMF) and solutions with different pH (pH = 5, pH = 9, pH = 11) (Figs. S9–S11).

### Adsorption and separation performances

The single-component adsorption isotherms of $C_3H_4$, $C_3H_4$(PD), $C_3H_6$, and $C_3H_8$ were measured to explore the adsorption performance of ZU-921 to ZU-924 and PCP-BDC (Figs. 3a, b and S12–S17). As depicted in Fig. 3a, b, all the $C_3H_8$, $C_3H_4$ (PD), and $C_3H_4$ adsorption capacities of F-functional ZU-921 are higher than that of $C_3H_6$ during the whole pressure range (0–1.0 bar), suggesting its preferential adsorption behavior towards $C_3H_4$, $C_3H_4$(PD), and $C_3H_8$ over $C_3H_6$. To our knowledge, this adsorption behavior has not been observed in the reported literature. The ideal adsorbed solution theory (IAST) method is applied to further evaluate the separation potentials of porous materials[6]. Considering the impurity content of $C_3H_4$, $C_3H_4$ (PD), and $C_3H_8$ in cracking gas typically account for 0.5–1%, 0.5–1% and 2–3%[29,34,47], respectively, the selectivity of $C_3H_8/C_3H_6$ (3 v/97 v), $C_3H_4/C_3H_6$ (1 v/99 v) and $C_3H_4$(PD)/$C_3H_6$ (1 v/99 v) binary mixtures on ZU-921 to ZU-924 and PCP-BDC were calculated (Fig. S18). As shown in Fig. 3c, ZU-921 exhibited high IAST selectivity for all of $C_3H_8/C_3H_6$ (3 v/97 v), $C_3H_4/C_3H_6$ (1 v/99 v), and $C_3H_4$(PD)/$C_3H_6$ (1 v/99 v) binary mixture at 298 K and 1.0 bar, which is up to 2.03, 2.17, and 2.03, respectively. Relatively, ZU-922 and PCP-BDC with aromatic-based pore environment exhibited moderate $C_3H_8/C_3H_6$ selectivity (1.49 and 1.75) but failed to selectively capture polar $C_3H_4$ and $C_3H_4$ (PD). ZU-924 with a high density of polar F-functional sites shows the priority adsorption sequence of $C_3H_4 > C_3H_4$ (PD) $> C_3H_6 > C_3H_8$ (Figs. 3d and S18). The experimental results demonstrate that the introduced density of the polar F-functional group is critical to afford the ideal materials for one-step $C_3H_6$ purification. The time-dependent adsorption curves demonstrate that the adsorption rates of all four C3 gases in ZU-921 are

close with no obvious kinetic effect, indicating that the good $C_3H_6$ purification performance of ZU-921 is governed by the thermodynamic equilibrium mechanism (Figs. S20–S22).

Furthermore, we conducted a detailed comparison study about the adsorption separation performance for four C3 gases of the literature-reported benchmark materials, for example, the $C_3H_4$ and $C_3H_4$(PD)-selective materials (NKMOF-1-Ni[30], NbOFFIVE-2-Cu-i[29], SIFSIX-3-Ni[24], and ZU-33 (GeFSIX-14-Cu-i)[47]) and $C_3H_8$-selective materials (Ni(ADC)(TED)$_{0.5}$[48] and FDMOF-2[34]). The $C_3H_4$, $C_3H_4$ (PD), $C_3H_6$, and $C_3H_8$ adsorption isotherms of these materials and their corresponding IAST selectivity of binary mixtures were measured and calculated (Figs. S23–S29 and Tables S4 and S5). None of these materials are able to simultaneously capture $C_3H_4$, $C_3H_4$ (PD), and $C_3H_8$ from $C_3H_6$ mixtures (Fig. 3e). In detail, the $C_3H_4$ and $C_3H_4$(PD)-selective materials with polar sites exhibit outstanding selectivities of $C_3H_4/C_3H_6$ and $C_3H_4$(PD)/$C_3H_6$, but the $C_3H_8/C_3H_6$ selectivity is lower than 1.0, indicating that the polar sites used to capture polar alkyne and allene are not beneficial for the alkane-selective adsorption. Similarly, the $C_3H_8$-selective materials with high-density weak interaction sites fail to capture $C_3H_4$ and $C_3H_4$(PD), and the $C_3H_8/C_3H_6$ selectivity is always low (≤2.0), revealing that the inert pore environment designed for the $C_3H_8$-selective adsorption could not be adapted for the accommodation of polar alkyne and allene. These results indicated that it was of great challenge to construct advanced porous materials that could selectively adsorb polar alkyne and allene, and inert alkane over alkene.

### Transient breakthrough experiments

Inspired by the unique adsorption behavior and high separation selectivity for C3 binary mixtures of ZU-921, the dynamic breakthrough experiments with mimicking C3 quaternary mixture proportions of

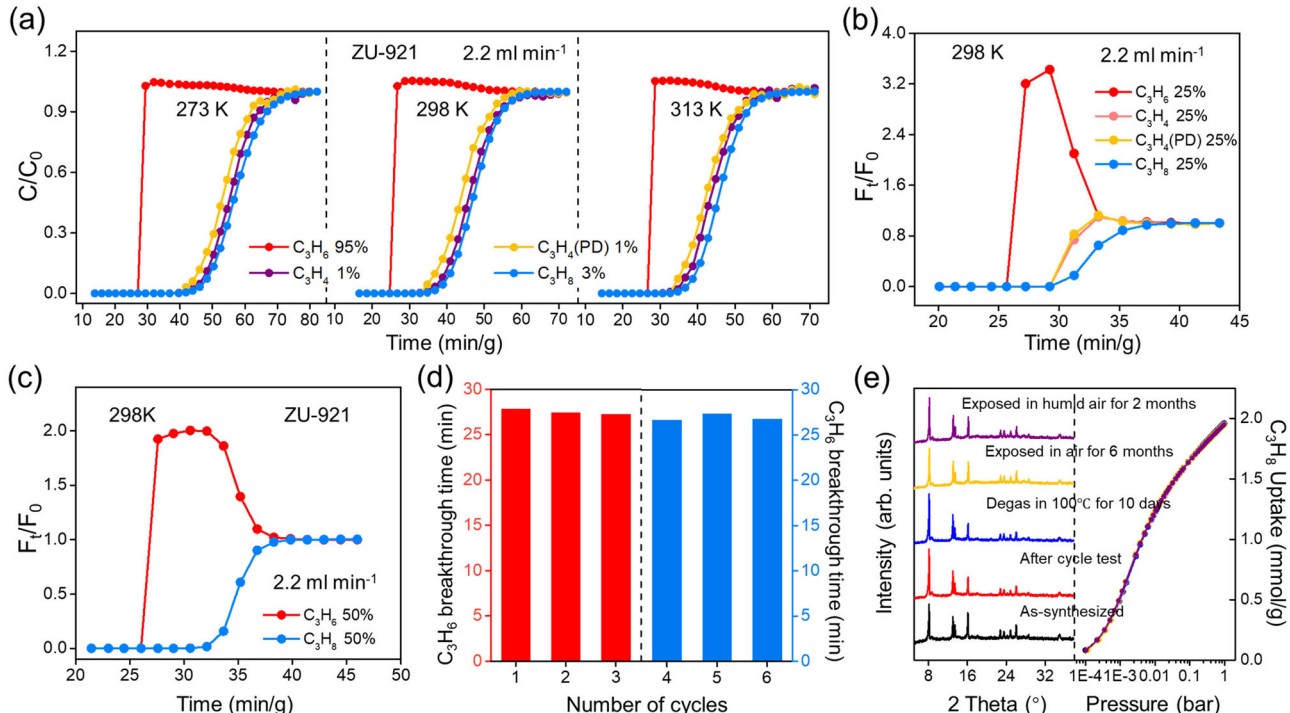

**Fig. 4 | Olefin purification.** Dynamic breakthrough curves of ZU-921 for (**a**) $C_3H_4$/$C_3H_4$(PD)/$C_3H_8$/$C_3H_6$ (1/1/3/95 $v/v/v/v$) mixture in $C_t$/$C_0$ under 273, 298 K and 313 K and 1.0 bar; **b** $C_3H_4$/$C_3H_4$(PD)/$C_3H_8$/$C_3H_6$ (25/25/25/25 $v/v/v/v$) mixture in $F_t$/$F_0$ under 298 K and 1.0 bar; **c** $C_3H_8$/$C_3H_6$ (50/50 $v/v$) mixture in $F_t$/$F_0$ under 298 K and 1.0 bar; **d** Six recycling breakthrough tests for $C_3H_8$/$C_3H_6$ (50/50 $v/v$, red) and $C_3H_4$/ $C_3H_4$(PD)/$C_3H_8$/$C_3H_6$ (1/1/3/95 $v/v/v/v$, light blue) separation with ZU-921 under 298 K and 1.0 bar; **e** The PXRD patterns and $C_3H_8$ adsorption isotherms of ZU-921 after different treatments. (column: 0.46 cm × 15 cm, 1.33 g or 0.46 cm × 25 cm, 2.28 g, flow rate: 2.2 mL min⁻¹). Source data are provided as a Source Data file.

cracking gas ($C_3H_4$/$C_3H_4$(PD)/$C_3H_8$/$C_3H_6$ 1 v/1 v/3 v/95 v) were conducted to evaluate its actual separation ability. The detailed experiment conditions and calculation methods of $C_3H_6$ productivity were described in the supporting information (Figs. S30–S31 and Table S13). As described in Fig. 4a, as the $C_3H_4$/$C_3H_4$(PD)/$C_3H_8$/$C_3H_6$ (1 v/1 v/3 v/ 95 v) mixture flowed through the column packed with ZU-921 under different temperatures (273 K, 298 K and 313 K), the $C_3H_6$ always eluted first, and then $C_3H_4$, $C_3H_4$(PD) and $C_3H_8$ broke out simultaneously, indicating good one-pot $C_3H_6$ purification performances of ZU-921. Specifically, 15.21 L/kg (99.99%) of $C_3H_6$ could be produced directly at 298 K and 1.0 bar (Fig. S32). The simple $C_3H_6$ purification process provides a potential energy-saving route to replace the current complex cascade purification ways. In addition, the separation performance towards equimolar C3 quaternary mixture ($C_3H_4$/$C_3H_4$(PD)/ $C_3H_8$/$C_3H_6$ 25/25/25/25 v/v/v/v) on ZU-921 was further explored, and the $C_3H_6$ still firstly eluted at the time of 26.5 min/g with the $C_3H_6$ (99.5%) productivity of 3.63 L/kg (Figs. 4b and S33 and S34). The roll-up phenomenon of $C_3H_6$ indicates that ZU-921 shows the weakest affinity towards $C_3H_6$, and all $C_3H_4$, $C_3H_4$(PD), and $C_3H_8$ were almost simultaneously eluted, demonstrating their close binding affinity within ZU-921 (Fig. 4b). We also explored the dynamic breakthrough performance of ZU-922 and FDMOF-2 under the same conditions. As shown in Figs. S35 and S36, $C_3H_6$, $C_3H_4$, and $C_3H_4$(PD) were simultaneously eluted, while $C_3H_8$, with the highest adsorption affinity, was well adsorbed. The results showed that ZU-922 and FDMOF-2 could only realize the selective $C_3H_8$ capture from $C_3H_6$, consistent with the adsorption isotherm results. Considering the complex competitive behavior in multi-component separations, the breakthrough experiments for binary mixtures of $C_3H_8$/$C_3H_6$ and $C_3H_4$/$C_3H_6$ were conducted (Figs. S37 and S38). As revealed by the breakthrough experiments of $C_3H_8$/$C_3H_6$ and $C_3H_4$/$C_3H_6$ binary mixtures, $C_3H_6$ is

always first eluted, followed by $C_3H_8$ or $C_3H_4$, indicating that ZU-921 could selectively capture the $C_3H_4$ and $C_3H_8$ to produce $C_3H_6$ directly. The $C_3H_6$ productivity of ZU-921 is up to 10.08 L/kg for the $C_3H_8$/$C_3H_6$ (50/50) binary mixture (Figs. 4c and S40), lower than FDMOF-2 (21.0 L/ kg)[34], but exceeding most of the reported materials, BUT-10 (3.95 L/ kg)[43] and WOFOUR-1-Ni (3.50 L/kg)[44] under the same conditions (Fig. S41). Moreover, the breakthrough experiments of $C_2H_2$/$C_2H_4$ (1/ 99) and $C_2H_6$/$C_2H_4$ (50/50) mixtures indicate that ZU-921 shows good $C_2H_6$-selective adsorptive performance, but poor $C_2H_2$/$C_2H_4$ separation performance (Fig. S43).

Given the importance of the recyclability and stability of porous materials for practical applications, the water and thermal stability of ZU-921 were investigated. Even under high humid conditions up to RH = 75%, the breakthrough performance of ZU-921 was invariable (Fig. S39). Meanwhile, the separation performance of ZU-921 was well maintained during the six cycling tests for $C_3H_4$/$C_3H_4$(PD)/$C_3H_8$/$C_3H_6$ and $C_3H_8$/$C_3H_6$ mixtures (Figs. 4d and S44–S46). No obvious structure degradation of ZU-921 was observed after it was exposed to different harsh conditions, such as humid air and high temperature (100 °C), as demonstrated by the invariant PXRD patterns and $C_3H_8$ adsorption isotherms (Fig. 4e). The impressive separation performance and the good stability of ZU-921 rendered it a promising adsorbent for one-step $C_3H_6$ purification from complex gas mixtures.

**10-times scale-up breakthrough experiment**
To further evaluate the application potential of ZU-921, we attempted the scale-up synthesis of ZU-921 and conducted the breakthrough experiments using the 10-times scale-up column (1.0 cm × 50 cm, 20.5 g) (Fig. S47). ZU-921 still exhibits impressive purification performance for different $C_3H_4$/$C_3H_4$(PD)/$C_3H_8$/$C_3H_6$ mixtures (1/1/3/95 $v/v/ v/v$ and 25/25/25/25 $v/v/v/v$) under different gas velocity (Figs. S48 and

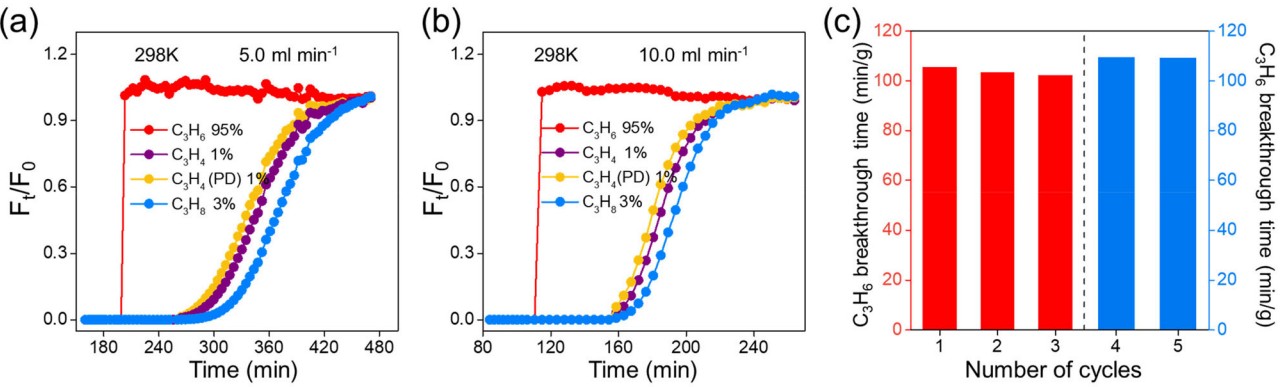

**Fig. 5 | 10-times scale-up breakthrough experiment.** Dynamic breakthrough curves of ZU-921 for C$_3$H$_4$/C$_3$H$_4$(PD)/C$_3$H$_8$/C$_3$H$_6$ (1/1/3/95 $v/v/v/v$) mixture in F$_t$/F$_0$ at **a** 5.0 ml min$^{-1}$, **b** 10.0 ml min$^{-1}$, and **c** the recycling tests for C$_3$H$_4$/C$_3$H$_4$(PD)/C$_3$H$_8$/ C$_3$H$_6$ (1/1/3/95 $v/v/v/v$, red; 25/25/25/25 $v/v/v/v$, light blue) after the regeneration under 393 K with the N$_2$ flow rate of 20.0 mL min$^{-1}$. (column: 1.0 cm × 50 cm, 20.5 g). Source data are provided as a Source Data file.

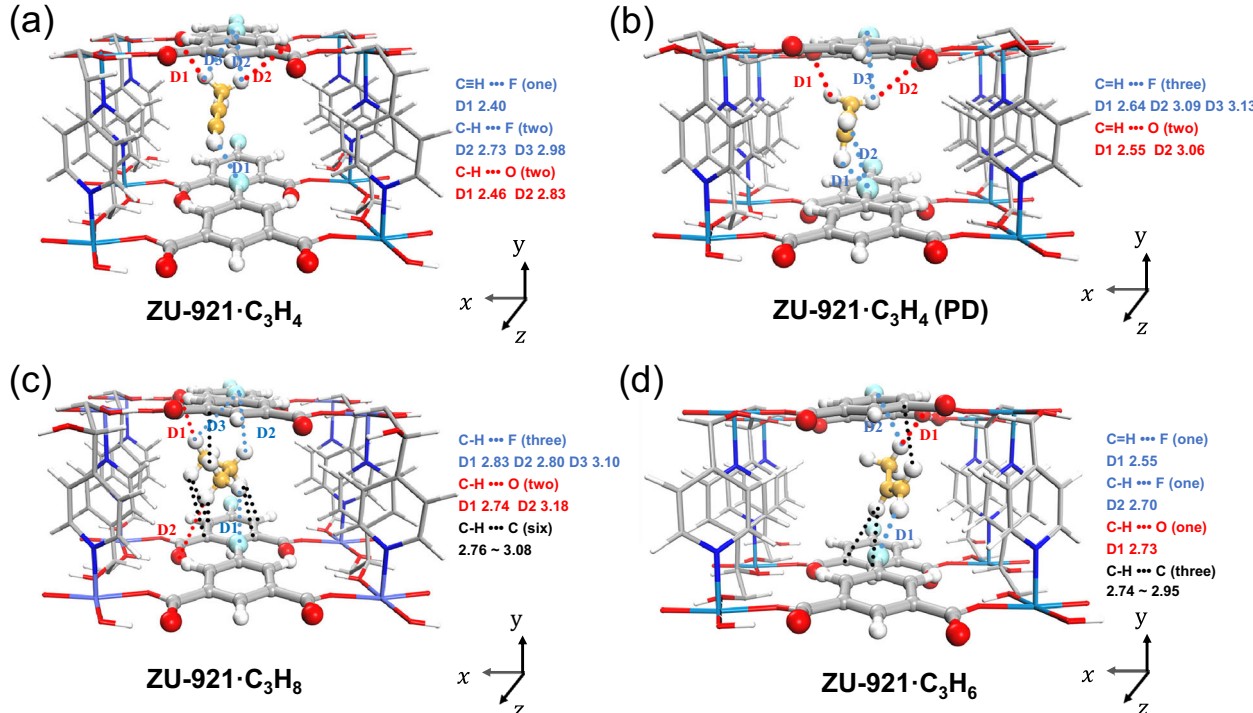

**Fig. 6 | DFT-D calculated binding sites for C3 gases in ZU-921.** **a** C$_3$H$_4$, **b** C$_3$H$_4$ (PD), **c** C$_3$H$_8$ and **d** C$_3$H$_6$ binding sites in ZU-921. The close contacts between framework atoms and the gas molecules are defined by the distances (in Å). (Framework: C, gray-80%; H, white; N, blue; O, red; Co, light blue; Gas: C, orange; H, white).

S49). The calculated C$_3$H$_6$ (99.99%) productivity is around 16.95 L/kg (5.0 mL/min) and 17.27 L/kg (10.0 mL/min) for the C$_3$H$_4$/C$_3$H$_4$(PD)/ C$_3$H$_8$/C$_3$H$_6$ (1/1/3/95 $v/v/v/v$) mixture (Figs. 5a, b and S50–S52). The column could be regenerated with nitrogen purge at 393 K for 800 min, and during the five consecutive cycling breakthrough experiments, the separation performance of ZU-921 remained invariable (Figs. 5c, S53–S56). We also evaluated the theoretical energy consumption for the C$_3$H$_6$ production, and the value of the adsorption process is lower than the cascade catalytic hydrogenation and distillation process used in industry (Tables S10–S12).

## Modeling simulation studies

To reveal the molecular-level adsorption behavior of C3 gases within the channel of ZU-921, we performed detailed modeling studies using the first-principles dispersion-corrected density functional theory

(DFT-D) method[45]. As shown in Figs. 6 and S57, all the C3 gases prefer to adsorb along the extending direction of the channel to get enough interactions with the parallel-arranged 5-fluoroisophthalic acid. In detail, C$_3$H$_4$ and C$_3$H$_4$(PD) are adsorbed in almost the same location. Each C$_3$H$_4$ molecule could interact with two negative fluorine atoms and two uncoordinated oxygen atoms via multiple cooperative hydrogen bonds (two C–H•••O (2.46 Å and 2.83 Å), two C–H•••F (2.73 Å and 2.98 Å) and C≡C–H•••F (2.40 Å)) (Fig. 6a). The C$_3$H$_4$(PD) molecule is bounded by two C=C–H•••O (2.55 Å and 3.06 Å) and three C=C–H••• F (2.64 Å, 3.09 Å and 3.13 Å) (Fig. 6b). While the C$_3$H$_8$ is inclined to interact with the paralleled three 5-fluoroisophthalic acid units via multiple van der Waals forces (six C–H•••C 2.76–3.08 Å) and multiple H-bonding interactions (C–H•••O 2.74 and 3.18 Å, C–H•••F 2.80, 2.83 and 3.10 Å) (Fig. 6c). In contrast, the interactions between ZU-921 and C$_3$H$_6$ are only provided by one C–H•••O (2.73 Å) and two

C−H•••F (2.55 Å and 2.70 Å) and three C−H•••C (2.74−2.95 Å) interactions (Fig. 6d). The binding energies of C3 gases on ZU-921 follow the sequence of $C_3H_8$ (−56.43 kJ/mol) > $C_3H_4$(PD) (−55.20 kJ/mol)≈$C_3H_4$ (−54.64 kJ/mol) > $C_3H_6$(−52.96 kJ/mol), which is consistent with the calculated $Q_{st}$ values of C3 gases based on their adsorption isotherms (Fig. S19). The results confirmed that ZU-921 could form a higher affinity with $C_3H_8$, $C_3H_4$, and $C_3H_4$ (PD) than $C_3H_6$. Simulation studies reveal that the polar fluorine, aromaticity sites, and uncoordinated oxygen are the keys to forming a synergistic binding environment for the simultaneous recognition of $C_3H_4$, $C_3H_4$ (PD), and $C_3H_8$ gases with different properties (Fig. S58).

## Discussion

In summary, we demonstrate the selective capture ability of ultra-microporous adsorbent ZU-921 for alkane, allene, and alkyne, achieving one-step $C_3H_6$ purification from C3 quaternary mixtures. The above remarkable progress in multiple impurities capture is attributed to the well-defined pore chemistry via ligand engineering strategy, the integrated binding sites of orderly lined aromatic units, as well as the quantified density of polar fluorine functional sites, enabling the exquisite control of priority affinity sequence of different molecules. Benefiting from the rational binding sequence, high-purity $C_3H_6$ (99.9% or 99.99%) could be easily obtained via a simple adsorption-desorption process from complex C3 mixtures. Our work not only presents an effective method to design advanced adsorbents for the simultaneous removal of multiple impurities but also demonstrates the great potential of adsorptive separation with tailor-made porous materials to simplify the complex separation process.

## Methods

### Chemicals

All reagents were analytical grade and used as received without further purification. $Co(NO_3)_2 \cdot 6H_2O$, $Zn(NO_3)_2 \cdot 6H_2O$, isophthalic acid (IPA), 5-fluoroisophthalic acid (IPA-F), 5-methylisophthalic acid (IPA-CH$_3$), terephthalic acid (BDC), 2,5-difluoroterephthalic acid (BDC-2F) and tetrafluoroterephthalic acid (BDC-4F) methanol (MeOH), and dimethylformamide (DMF) were purchased from Aladdin Reagent Co. Ltd., Meso-α,β-Di(4-pyridyl) Glycol (DPG) was purchased from TCI Co. Ltd. 2,5-bis(trifluoromethyl) terephthalic acid (BDC-(CF$_3$)$_2$) and 1,4-diazabicyclo [2.2.2] octane (DABCO) were purchased from Aladdin. Ultrahigh purity grade He (99.999%), $N_2$ (99.999%), $C_3H_4$ (99.9%), $C_3H_4$ (PD, 99.9%), $C_3H_6$ (99.99%), $C_3H_8$ (99.99%), and mixed gas ($C_3H_4$/$C_3H_6$ = 1/99, $v/v$, $C_3H_6$/$C_3H_8$ = 50/50, $v/v$, $C_3H_4$/$C_3H_4$(PD)/$C_3H_8$/$C_3H_6$ = 1/1/3/95, $v/v/v/v$, $C_3H_4$/$C_3H_4$(PD)/$C_3H_8$/$C_3H_6$ = 25/25/25/25, $v/v/v/v$) were purchased from Shanghai Wetry Standard gas Co., Ltd. (China) and used for all measurements.

### Material synthesis

PCP-IPA-X[33,49] and the comparison materials of FDMOF-2[34], Ni(ADC)(TED)$_{0.5}$[48], NKMOF-1-Ni[30], NbOFFIVE-2-Cu-i[29], SIFSIX-3-Ni[24], and ZU-33[47] were synthesized according to the previously reported procedure.

### ZU-921 (PCP-IPA-F)

81 mg DPG was dissolved in DMF/MeOH (1:1, 30 mL) at 60 °C, and 75 mg IPA-F and 109 mg $Co(NO_3)_2 \cdot 6H_2O$ were dissolved in 5 mL MeOH. Then, the two solutions were mixed and heated at 80 °C for 24 h to yield as-synthesized ZU-921, with the yield reaching up to 82% (based on DPG ligand).

### Scale-up preparation of ZU-921

5.0 g DPG was dissolved in DMF/MeOH (1:1, 1.5 L) at 60 °C, and 4.63 g IPA-F and 6.7 g $Co(NO_3)_2 \cdot 6H_2O$ were dissolved in 50 mL MeOH. Then, the two solutions were mixed and heated at 80 °C for 24 h to yield as-synthesized ZU-921, with the yield reaching up to 80% (based on DPG ligand).

### ZU-922 (PCP-IPA-CH$_3$)

81 mg DPG was dissolved in DMF/MeOH (1:1, 30 mL) at 60 °C, and 75 mg IPA-CH$_3$ and 109 mg $Co(NO_3)_2 \cdot 6H_2O$ were dissolved in 5 mL MeOH. Then, the two solutions were mixed and heated at 80 °C for 24 h to yield as-synthesized ZU-922, with the yield reaching up to 80% (based on DPG ligand).

### ZU-923 (PCP-BDC-2F)

81 mg DPG was dissolved in DMF/MeOH (1:1, 30 mL) at 60 °C, and 80 mg BDC-2F and 109 mg $Co(NO_3)_2 \cdot 6H_2O$ were dissolved in 5 mL MeOH. Then, the two solutions were mixed and heated at 80 °C for 24 h to yield as-synthesized ZU-923, with the yield reaching up to 77% (based on DPG ligand).

### ZU-924 (PCP-BDC-4F)

81 mg DPG was dissolved in DMF/MeOH (1:1, 30 mL) at 60 °C, and 95 mg BDC-4F and 109 mg $Co(NO_3)_2 \cdot 6H_2O$ were dissolved in 5 mL MeOH. Then, the two solutions were mixed and heated at 80 °C for 24 h to yield as-synthesized ZU-924, with the yield reaching up to 75% (based on DPG ligand).

### FDMOF-2

A mixture of $Zn(NO_3)_2 \cdot 6H_2O$ (0.4 mmol), the BDC-(CF$_3$)$_2$ (0.4 mmol) and DABCO (0.2 mmol), DMF (15 mL), and two drops of HNO$_3$ were added into a 25 mL glass vial. After the mixture was stirred for 30 min, the vial was sealed and heated at 120 °C for 48 h to yield as-synthesized FDMOF-2 with the yield reaching up to 65% (based on BDC-(CF$_3$)$_2$ ligand).

### Sample characterization

Powder X-ray diffraction (PXRD) patterns were collected using a PANalytical Empyrean series 2 diffractometer with Cu-Ka radiation, at room temperature, with a step size of 0.0167°, a scan time of 15 s per step, and 2θ ranging from 5 to 50°. The thermogravimetric analysis (TGA) data were collected in a NETZSCH Thermogravimetric Analyzer (STA2500) from 50 to 700 °C with a heating rate of 10 °C/min. The morphology was investigated using a NOVA 200 Nanolab scanning electron microscope (SEM). The 195 K $CO_2$ and 77 K $N_2$ adsorption/desorption isotherms were obtained on a Micromeritics 3Flex and BSD-660 volumetric adsorption apparatus. The apparent Langmuir surface area was calculated using the adsorption branch with the relative pressure $P/P_0$ in the range of 0.005−0.1. The total pore volume ($V_{tot}$) was calculated based on the adsorbed amount of $CO_2$ or $N_2$ at the $P/P_0$ of 0.99. The pore size distribution (PSD) was calculated using the H-K methodology with $CO_2$ adsorption isotherm data and assuming a slit pore model.

### Gas adsorption measurements

The $C_3H_4$, $C_3H_4$(PD), $C_3H_6$, and $C_3H_8$ adsorption-desorption isotherms at different temperatures were measured volumetrically by Micromeritics 3Flex and BSD-660 adsorption apparatus for pressures up to 1.0 bar. Prior to the adsorption measurements, the samples were degassed using a high vacuum pump (<5 µm Hg) at 373 K for over 12 h.

### Kinetic adsorption measurement

The time-dependent adsorption profiles of $C_3H_4$, $C_3H_4$(PD), $C_3H_8$, and $C_3H_6$ were measured using an Intelligent Gravimetric Analyzer (IGA-100, HIDEN, U.K.). The diffusional time constants (D′, D/r²) were calculated by the short-time solution of the diffusion equation assuming a step change in the gas-phase concentration, clean beds initially, and micropore diffusion control:

$$\frac{M_t}{M_e} = \frac{6}{\sqrt{\pi}} \cdot \sqrt{\frac{D}{r^2}} \cdot \sqrt{t}$$

Where t (s) is the time, $M_t$ (mmol/g) is the gas uptake at time t, $M_e$ is the gas uptake at equilibrium (mmol/g), D ($m^2 s^{-1}$) is the diffusivity, and r (m) is the radius of the equivalent spherical particle. The slopes of $M_t$/$M_e$ versus $\sqrt{t}$ are derived from the fitting of the plots at 298 K and different adsorption pressures.

## Breakthrough experimental

The breakthrough experiments were carried out in a homemade apparatus under a standard procedure. First, the samples were degassed under vacuum at 393 K for 12 h and then were introduced to the adsorption column with different sizes (0.46 cm × 15 cm or 0.46 cm × 25 cm, or 1.0 cm × 50 cm). Second, the adsorption column was connected to the homemade apparatus, and the carrier gas (He ≥99.999%) purged the adsorption column for more than 1 h to ensure that the adsorption bed was clean and saturated with He. Third, we switched the carrier gas to the desired gas mixture without any inert gas dilution ($C_3H_4$/$C_3H_6$ = 1/99, $v/v$, $C_3H_6$/$C_3H_8$ = 50/50, $v/v$, $C_3H_4$/$C_3H_4$(PD)/$C_3H_8$/$C_3H_6$ = 1/1/3/95 $v/v/v/v$, $C_3H_4$/$C_3H_4$(PD)/$C_3H_8$/$C_3H_6$ = 25/25/25/25 $v/v/v/v$). Fourth, the desired gas flows through the column until the concentrations of all the components are consistent with the entrance of the gas mixture (temperature: 298 K, pressure: 1.0 bar). In this process, the eluted gas was passed to an analyzer port and analyzed using gas chromatography (GC490 Agilent) with a thermal conductivity detector (TCD), or gas chromatography (Shimadzu GC2010) with a flame ionization detector (FID). To obtain the breakthrough curves in $F_t$/$F_0$ ($F_t$ and $F_0$ are the flow rates of each gas at the outlet and inlet, respectively), the gas chromatography (Shimadzu GC2010) with a flame ionization detector (FID) is employed, and the loop capacity of the gas chromatography is 5 mL, and its injection time is 0.3 min. Within the 0–10 mL/min range of inlet gas flow rate ($F_0$), the injection volume would not fill up the capacity of the loop, allowing the outlet gas flow rate ($F_t$) of each gas to be determined from the peak area. After the breakthrough experiment, the adsorption column was regenerated at 393 K or 423 K with a 20 mL/min $N_2$ flow rate for 10–12 h. Detailed experiment parameters like flow rates, temperatures, and column sizes are provided in every breakthrough curve.

## Isotherm fitting

The pure-component isotherms of $C_3H_4$, $C_3H_4$(PD), $C_3H_6$, and $C_3H_8$ were fitted using a two-site Langmuir-Freundlich model for the full range of pressure (0-1.0 bar).

$$q = q_{sat1}\frac{b_1 p^{v1}}{1 + b_1 p^{v1}} + q_{sat2}\frac{b_2 p^{v2}}{1 + b_2 p^{v2}} \qquad (1)$$

Here, p is the pressure of the bulk gas at equilibrium with the adsorbed phase (bar), q is the adsorbed amount per mass of adsorbent (mmol $g^{-1}$), $q_{sat}$ is the saturation capacity (mmol $g^{-1}$), b is the affinity coefficient (bar$^{-1}$), and v represents the deviation from an ideal homogeneous surface.

## Isosteric heat of adsorption

The isosteric heat of $C_3H_4$, $C_3H_4$(PD), $C_3H_6$, and $C_3H_8$ adsorption, $Q_{st}$, defined as

$$Q_{st} = RT^2\left(\frac{\partial lnP}{\partial T}\right)_q \qquad (2)$$

were determined using the pure-component isotherm fits using the Clausius-Clapeyron equation. where $Q_{st}$ (kJ/mol) is the isosteric heat of adsorption, T (K) is the temperature, P (bar) is the pressure, R is the gas constant, and q (mmol/g) is the adsorbed amount.

## IAST calculations

The selectivity of the preferential adsorption of component 1 over component 2 in a mixture containing 1 and 2 can be formally defined as:

$$S = \frac{x_1/y_1}{x_2/y_2} \qquad (3)$$

In the above equation, $x_1$ and $y_1$ ($x_2$ and $y_2$) are the molar fractions of component 1 (component 2) in the adsorbed and bulk phases, respectively. We calculated the values of $x_1$ and $x_2$ using the Ideal Adsorbed Solution Theory (IAST) of Myers and Prausnitz[50].

## Density functional theory calculations

First-principles density functional theory (DFT) calculations were performed using Materials Studio's CASTEP code[45]. All calculations were conducted under the generalized gradient approximation (GGA) with Perdew−Burke−Ernzerhof (PBE). A semiempirical addition of dispersive forces to conventional DFT was included in the calculation to account for van der Waals interactions. Cutoff energy of 544 eV and a 2 × 2 × 3 k-point mesh was found to be enough for the total energy to be covered within 0.01 meV atom$^{-1}$. The structures of the synthesized materials were first optimized from the refined structures. To obtain the binding energy, the pristine structure and an isolated gas molecule placed in a supercell (with the same cell dimensions as the pristine crystal structure) were optimized and relaxed as references. $C_3H_4$, $C_3H_4$(PD), $C_3H_6$, and $C_3H_8$ gas molecules were then introduced to different locations of the channel pore, followed by full structural relaxation. The static binding energy was calculated by the equation:

$$E_B = E(gas) + E(adsorbent) - E(adsorbent + gas) \qquad (4)$$

## Structure simulation

The high-quality Powder X-ray diffraction (PXRD) data for Rietveld refinement of ZU-921 and ZU-922 frameworks were collected using a PANalytical Empyrean series 2 diffractometer with Cu-Ka radiation, at room temperature, with a step size of 0.01313°, a scan time of 198.65 s per step, and 2θ ranging from 5 to 90°. As indicated by the similar PXRD patterns and assembled modules, the structure of ZU-921 and ZU-922 is speculated to be isostructural to PCP-IPA except for the different substituted groups (−F, −$CH_3$, and −H) (Fig. S2). We built the initial raw structure of ZU-921 and ZU-922 based on the framework of PCP-IPA using Material Studio. The Rietveld refinement, a software package for crystal determination from the XRD pattern, was performed to optimize the lattice parameters iteratively until the $w$Rp value converges. The Reflex Powder Solve was employed to further optimize the atomic positions, bond length, bond angle, etc. The pseudo-Voigt profile function was used for whole profile fitting, and the Berrar-Baldinozzi function was used for asymmetry correction during the refinement processes. Line broadening from crystallite size and lattice strain was considered.

## Data availability

Crystallographic data for the structures reported in this article have been deposited at the Cambridge Crystallographic Data Centre, under deposition numbers CCDC 2294207 (for ZU-921) and 2294206 (for ZU-922). Copies of the data can be obtained free of charge via https://www.ccdc.cam.ac.uk/structures/. The authors declare that the main data supporting the findings of this study are available within the article and its Supplementary Information files. Source data that support the findings of this study are provided as a source data file. Source data are provided with this paper.

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

## Acknowledgements

The work was supported financially by the National Natural Science Foundation of China (No. 22122811 and 22438011), the Natural Science Foundation of Zhejiang Province (No. LD25B060002), the "Pioneer" and "Leading Goose" R&D Program of Zhejiang (No. 2024C01202), and the Research Computing Center in the College of Chemical and Biological Engineering at Zhejiang University.

## Author contributions

H.X., L.Y., and P.Z. conceived the project idea, designed the research, and co-wrote the manuscript. P.Z. carried out the materials synthesis, adsorption experiments, dynamic breakthrough measurements, and computational simulation. Z.Q. performed the IAST calculation, the material structure characterization, and stability testing. X.S. and X.C. conducted the cycling breakthrough experiment. Y.L. and S.C. performed the energy consumption calculation and the 10-times scale-up preparation of materials. All authors contributed to the discussion of results and commented on the manuscript.

## Competing interests

The authors declare no competing interests.
