## [Transparent Peer Review file · Nature Communications]

One-step Propylene Purification from a Quaternary Mixture by a Single Physisorbent

Corresponding Author: Professor Huabin Xing

Version 0:

Reviewer comments:

Reviewer #1

(Remarks to the Author)

The authors have sufficiently addressed the reviewer's comments and revised the manuscript accordingly. Therefore, I recommend the acceptance of this manuscript. One minor concern is that the authors repeatedly state that this is the first demonstration or first example of achieving one-step C₃H₆ purification from C₃ quaternary mixtures. Please note that last year, a paper published in JACS (J. Am. Chem. Soc. 2024, 146, 44, 30155–30163) reported one-step C₃H₆ purification from C₃ quaternary mixtures with MOF materials.

Dear reviewers,

Oct 23, 2025

We greatly appreciate the positive feedback and for recommending acceptance from you, and we have revised the manuscript and supplementary information accordingly as detailed in the responses below.

Comments: *The authors have sufficiently addressed the reviewer's comments and revised the manuscript accordingly. Therefore, I recommend the acceptance of this manuscript. One minor concern is that the authors repeatedly state that this is the first demonstration or first example of achieving one-step C₃H₆ purification from C₃ quaternary mixtures. Please note that last year, a paper published in JACS (J. Am. Chem. Soc. 2024, 146, 44, 30155 – 30163) reported one-step C₃H₆ purification from C₃ quaternary mixtures with MOF materials.*

Response: Thank you for your positive feedback and for recommending acceptance. We also appreciate you pointing out the relevant reference (J. Am. Chem. Soc. 2024, 146, 44, 30155–30163). We have revised the manuscript accordingly and removed the claims of being the “first demonstration”. The detailed modifications are as follows.

Modifications: Manuscript: Page 1, 2 and 17

Abstract section: “ZU-921 is presented as the demonstration that solves the long-standing challenge in one-step propylene (C₃H₆) purification from the C₃ quaternary mixture.”

Introduction section: “However, despite the above progress, one-step C₃H₆ purification from quaternary C₃ mixtures containing C₃H₄, C₃H₄(PD), and C₃H₈ via a single physisorbent still remains a grand challenge³⁹.”

“Herein, we solved the challenge of one-step C₃H₆ purification from the quaternary mixtures using tailor-made ZU-921 (ZU-represents Zhejiang University) featuring orderly lined aromatic surfaces and optimal density of electronegative sites.”

Discussion: “In summary, we demonstrate the selective capture ability of ultramicroporous adsorbent ZU-921 for alkane, allene, and alkyne, achieving one-step C₃H₆ purification from C₃ quaternary mixtures.”

Reference: 39 Xie, X.-J. *et al.* Surface chemistry regulation in Cu₄I₄-triazolate metal–organic frameworks for one-step C₃H₆ purification from quaternary C₃ mixtures. *J. Am. Chem. Soc.* **146**, 30155–30163 (2024).

We believe that we have satisfactorily addressed reviewer’s concerns, and that the manuscript is thereby improved. We thank you in advance for your consideration of this revised manuscript.